# Preparatory Conditions Optimization and Characterization of Hierarchical Porous Carbon from Seaweed as Carbon-Precursor Using a Box—Behnken Design for Application of Supercapacitor

**DOI:** 10.3390/ma15165748

**Published:** 2022-08-20

**Authors:** Wein-Duo Yang, Jing-Xuan Wang, Yu-Tse Wu, Hsun-Shuo Chang, Horng-Huey Ko

**Affiliations:** 1Department of Chemical and Materials Engineering, National Kaohsiung University of Science and Technology, Kaohsiung City 807, Taiwan; 2School of Pharmacy, College of Pharmacy, Kaohsiung Medical University, Kaohsiung City 807, Taiwan

**Keywords:** seaweed, hierarchical, porous activated carbon, Box–Behnken design

## Abstract

This study has developed an environmentally friendly, simple, and economical process by utilizing seaweed as a carbon precursor to prepare a hierarchical porous carbon for the application of a supercapacitor. In the carbonization process, the design of experiment (DOE) technology is used to obtain the optimal preparatory conditions with the best electrochemical properties for the electrode materials of supercapacitors. Without using strong acid and alkali solution of the green process, NaCl is used as the pore structure proppant of seaweed (SW) for carbonization to obtain hierarchical porous carbon material to improve the pore size distribution and surface area of the material. In the experiment of SW activation, the interaction between factors has been explored by the response surface methodology (RSM) and Box–Behnken design, and the optimal conditions are found. The activated carbon with the specific surface area of 603.7 m^2^ g^−1^ and its capacitance reaching 110.8 F g^−1^ is successfully prepared. At a current density of 1 A g^−1^, the material still retains 95.4% of the initial capacitance after 10,000 cycles of stability testing. The hierarchical porous carbon material prepared by the design of experiment planning this green process has better energy storage properties than supercapacitors made of traditional carbon materials.

## 1. Introduction

Supercapacitors (SCs) possessing high power density with the advantages of short charging time and long service life have great development potential in energy applications [1]. Until now, carbon electrode materials [2,3,4,5] are one of the most important electrode materials for supercapacitors. Although carbon materials can be used to prepare supercapacitors with stable cycling properties and fast charge–discharge performance, they cannot be precisely controlled in terms of the optimal pore size distribution and specific surface area to affect the conductivity and capacitance of the electrode materials [6]. Specific surface area, pore structure, surface functional groups, and electrical conductivity are important properties that determine the capacitive performance of carbon materials [7]; however, the specific capacitance of activated carbon is relatively low, thus limiting the development of its applications.

Hierarchical porous carbon includes interconnected micro-pores, meso-pores, and macro-pores. The preparation of hierarchical porous carbon materials is often obtained by using template technology [8,9,10] and activation process [11,12]. Template technology requires a large number of inorganic or organic templates, and the preparation and removal of templates is complicated. In contrast, activation can provide porous carbon with high surface area, but using large amounts of activators, such as KOH [13], NaOH [14], H_3_PO_4_ [15], and ZnCl_2_ [16], etc., also requires subsequent treatment, and the generated acid and lye are harmful to the environment [17].

In addition, Huo et al. [18] utilized potassium thiocyanate (KSCN) as activator through one-step synthesis to obtain high-performance N/S co-doped porous carbon material for environmental remediation. The synthesis of hybrid materials consists of commercial Norit carbons and oligothiophenes. It was shown that the surface chemistry of activated carbon plays a key role in the dye deposition process [19]. Shrestha [20] reported that wood-dust of Dalbergia sisoo (Sisau) derived was activated with H_3_PO_4_ to obtain activated carbon and used as an adsorbent material for the removal of rhodamine B dye from an aqueous solution successfully. Recently, Wang et al. [21] reported a glucose-assisted hydrothermal method for directly transforming metal-organic frameworks (MOFs) into hollow carbonaceous materials. The as-obtained ZnO/C materials with hollow interiors exhibit more active sites, which are supported by their superior electrochemical performance for supercapacitor applications. A novel hierarchical porous carbon with a combination of highly conductive electronic pathways and rich ionic storage units in three-dimensional network morphology, leading to high specific capacitance of electric double-layer capacitor (EDLC) were studied. By facile hydrothermal synthesis and carbonization, the carbon electrode derived from MOF and polymer fibers, exhibits extremely high specific capacitance of ∼385 F g ^−1^ at 0.1 A g^−1^ and maintain capacitance of 303 F g^−1^ at 10 A g^−1^ [22]. Moreover, a solvent-assisted approach for preparing ultrathin nanosheet-assembled nickel-based metal–organic framework (Ni-MOF) microflowers with high specific capacitance and excellent rate capability as an electrode material for supercapacitors has been reported [23]. Particularly, a superior sodium metal anodes enabled by sodiophilic carbonized coconut framework with 3D tubular structure was reported. The oxygen functional groups with sodiophilicity contribute to the adsorption of Na^+^ and reduce the Na nucleation energy on the surface of O-CCF. The interaction of 3D tubular structure and oxygen functional groups enable Na stripping/plating over 10,000 cycles at 50 mA cm^−2^ [24]. Besides, Feng et al. [25] reviewed comprehensively summarizes recent advances in micro-supercapacitors (MSCs) for AC line-filtering, from fundamental mechanisms to appropriate characterization and emerging applications, as well as also presented perspectives and insights into the development of MSCs in different frequency ranges for their applications. In recent years, pomelo peel [26], rice husk [27], Chinese date [28], coconut shell [29], and other biomass have been used for the in situ synthesis of heteroatom self-doped carbon materials. Among these precursors, biomass has received extensive attention due to its abundant sources, low price, and environmental friendliness [30]. Particularly, seaweed (SW) is an oxygen-rich biomass [31] and has an inherent micro-pore structure, which is a good microstructure of precursor for the preparation of interconnected carbon materials. In today’s scarcity of resources, seaweed has gradually attracted the attention of all walks of life. Seaweed is a natural carbon source, and carbon precursors can be synthesized from seaweed by a simple thermal cracking method without complex activation treatment, and hierarchical porous carbon materials can be obtained. It is an ideal electrode material for supercapacitors.

It is possible to use a template or an activator in the biomass carbonization process to obtain activated carbon with a high specific surface area. However, subsequent treatment is required, and the produced acid and lye are harmful to the environment. If NaCl is used as a template [32,33], sodium chloride can be removed by washing with water only; in the carbonization process, the recrystallized NaCl has stable chemical properties, it is an environmentally friendly green process.

Zuo et al. [34] used a natural and cheap biomass waste—nitrogen-rich pruning of Lycium barbarum L.—as carbon precursors and used NaCl as an intercalating agent to prepare activated carbon materials. The obtained carbon material has porous structure, large surface area, increased N content in the microstructure and high graphitized carbon, so that it has high electrocatalytic activity for oxygen reduction reaction and is suitable for cathode electrocatalyst of fuel cell. In addition, Hencz [35] et al., firstly synthesized hierarchical porous carbon by pyrolysis of algal biomass under an inert atmosphere with NaCl as activator and used it as a cathode material for lithium-sulfur (Li-S) batteries. The algal carbon shows a microporous/mesoporous/macroporous structure with a high total pore volume of 1.48 cm^3^ g^−1^ and a high surface area of 1510.71 m^2^ g^−1^, which is favorable for accommodating a large amount of sulfur, confirming that the biomass-derived porous carbon, suitable for assembling high-performance Li-S batteries. Algae, a fibrous substance, has the unique property of producing high levels of oxygen after pyrolysis; oxygen surrounds the electrodes of carbon materials, allowing them to absorb ionic electrolytes more quickly. Xue et al. [36] impregnated seaweed in salt without the absence of acid to synthesize oxygen-rich activated carbon materials with unique pores, which can be utilized to assemble supercapacitors with excellent energy density. Sodium chloride can be dipped into seaweed for carbonization and can be removed by washing with only water. During the carbonization process, the recrystallized NaCl is chemically stable and acts as a pore proppant in seaweed, promoting to maintain the original pore structure, and as the template for the formed holes. NaCl can be completely removed by water washing, which is a very environmentally friendly process. Wang et al. [37] used water-soluble NaCl/FeCl_3_ as a hard template to controllably introduce mesoporous pores through a pyrolysis-washing process. Using nitrogen (N)-rich polyvinylpyrrolidone (PVP) as the carbon precursor, the as-obtained porous carbon materials have excellent properties with a pore size of 4 nm. Moreover, the electrode material exhibits a specific capacitance of 118 F g^−1^ in H_2_SO_4_ electrolyte. In the process, NaCl is easy to be recrystallized, precipitated and removed, and can be reused, making this method to prepare a very forward-looking hierarchical porous carbon material.

Design of experiment (DOE) is often used in the design of experiments and the significant factors of the preparatory conditions can be tested by analysis of variance. This is also a reliable method to simplify the process of identifying the most influential preparation variables, and the optimal preparatory conditions can be singled out [38,39]. Moreover, it explores the interaction between factors effectively and increases the accuracy of experimental results, reduce the number of experiments [40]. Response surface methodology (RSM) is firstly used to select the important factors from many factors, and then uses the first-order model of steepest ascent/descent to improve the speed and efficiency of the path. The ground is towards the vicinity of the optimal point; after reaching the vicinity of the optimal point, the Box–Behnken design is used for analysis to find the optimal conditions. Using Box–Behnken design, the variation of predicted values within the experimental design area is reasonably stable, and the reproducibility and stability of the predicted surface model are improved [41]. Box–Behnken design is a rotatable design that can efficiently estimate quadratic terms. In particular, NaCl is introduced as a template agent, and there is no need to use polluting and toxic chemicals in the process, which is very friendly to the environment. Therefore, combining with Box–Behnken design to optimize the process of hierarchical porous carbon production for application of supercapacitors from seaweed as carbon source has developing potential. In addition, to the best of our knowledge, the related research is very rare and worth to study.

In this study, seaweed was used as carbon precursor, water-soluble NaCl was the primary skeleton support of seaweed, and prepare a hierarchical microporous/mesoporous/macroporous porous carbon materials by carbonization route to enhance the electrode performance of supercapacitors. The preparatory conditions of the process were optimized. Furthermore, in the process of the porous carbon material without using strong acid and alkali, the carbon material obtained by freeze-drying technology greatly reduced the shrinkage and deformation of the carbon material during the drying process, and obtain an excellent pore properties of carbon material, which can greatly improve its electrochemical properties.

## 2. Experimental

### 2.1. Preparation of Hierarchical Porous Activated Carbon

Seaweed (from Kenting Beach, Ping-Tong, Southern Taiwan) was dried in an oven at 105 °C for 24 h, and pulverized into fine powder, then the powder was sieved. The powder between 20–100 mesh was taken and stored in a refrigerator at 4 °C.

Before carrying out the DOE experimental design, a preliminary study on the possible influencing variables must be made. The range of the variables to be tested is wider, and then narrowed down to the range that has a greater impact on the optimization results. Then, according to the results of preliminary study to carry out Box–Behnken design and optimize the process conditions.

A total of 10 g of dried seaweed powder was mixed with different weight ratios of NaCl at weight ratios of NaCl/seaweed being 1/1~5/1. Then, put the above mixture into a tube furnace under an environment with a N_2_ flow rate of 50 mL min^−1^ at temperature raising rate of 10 °C min^−1^ up to 700~900 °C and kept 1 h for high temperature carbonization. After cooling, the carbonized seaweed was washed with ID-water to remove NaCl and other impurities, and the pH-value was approximate 7, finally. A hierarchical porous carbon material was prepared by freezing dryer after drying for 48 h. The preparatory procedure of AC from seaweed is shown as Figure 1.

### 2.2. Design of Experiment (DOE) for Carbonization of Seaweed

The aim of optimization was to find the optimized conditions of the experimental variables of porous activated carbon by carbonization of seaweed. Design Expert (version 13) (Stat-Ease Inc., Minneapolis, MN, USA) software was utilized to design and analysis of the experiment. Based on the results obtained in preliminary experiments, the following four variables were chosen as the experimental variables: carbonization temperature (A), carbonization time (B), weight ratio of NaCl/seaweed (C), and weight ratio of water/seaweed (D). Hence, these variables were selected to find the optimized conditions for a higher specific capacitor of the as-obtained porous activated carbon electrode using response surface methodology (RSM) and Box–Behnken design. The range and levels of the experimental variables investigated in this study are given in Table 1. The central values (zero level) chosen for experimental design were carbonization temperature at 700 °C, carbonization time of 60 min, weight ratio of NaCl/seaweed at 4, and weight ratio of water/seaweed at 10. Using Design Expert to plan seaweed carbonized for preparation of hieratical porous activated carbon and use random sampling method to arrange the preparation sequence of the experiment (the column “Run No.” in Table 2). Table 2 shows the design of experiment (DOE) table and experimental result.

According to the principle of normal distribution, the individual measured data (X) must be within the standard deviations of the data (σ) at 1σ, 2σ, and 3σ, respectively, reaching at least 68.3%, 95.4%, and 99.7%. Thus, such experimental data can be considered accurate. The data (Cs in Table 2) obtained by measuring the GCD test 5 times for each of the 25 run of electrodes is fit with the above statistical principles, indicating that the capacitance obtained from the test has high accuracy. Moreover, the Cs of the sample is expressed as mean value and standard error of the mean (Appendix A).

### 2.3. Characterizations

A field-emission scanning electron microscope (FESEM) (JEOL, JEOL6330, Tokyo, Japan) with an acceleration voltage of 80 kV was utilized to examine the microstructures of the as-prepared micropore/mesopore/macropore hierarchical porous carbon materials. The Brunauer–Emmett–Teller (BET) surface area, Barrett–Joyner–Halenda (BJH) mesopore area, t-plot micropore area, and N_2_ adsorption–desorption isotherms were measured with a Micrometrics ASAP 2020 instrument (Micrometrics, Atlanta, GA, USA). The I_D_/I_G_ ratio of Raman spectra shift was determined using a Jobin-Yvon Lab Ram HR800 Raman spectroscope (HORIBA Jobin Yvon Inc., Paris, France) equipped with a 514.5 nm laser source.

### 2.4. Electrochemical Properties

The mixture of AC materials, polyvinylidene-fluoride (PVDF) as a binder, and black carbon as a conductive additive at a weight ratio of 85:10:5 was dispersed in *N*-methyl pyrrolidone and then mixed for 12 h by magnetic stirring to generate a homogeneous mixture. The resultant slurry was coated onto Ni foam as a current collector.

All electrochemical measurements were carried out in 1 M Na_2_SO_4_ electrolyte. The electrochemical experiment was measured in a conventional three-electrode system, using the as-prepared AC/Ni foam composite (1.0 cm × 1.0 cm) as the working electrode, a Pt wire (1.0 cm × 1.0 cm) as the counter electrode, and a saturated calomel electrode (SCE) as the reference electrode.

Cyclic voltammetry (CV) and galvanostatic charging–discharging (GCD) were measured using a CHI 760D electrochemical workstation.

Moreover, the specific capacitance (C_s_) was obtained according to the discharge curve of the GCD test using the following Equation (1):(1)$$C_{s}=\frac{i\times \Delta t}{\Delta V\times m}$$
where i (A) is the discharge current, Δt (s) is the discharge time, ΔV (V) is the discharging potential difference, and m (g) is the mass of the porous activated carbon material.

Electrochemical impedance spectra (EIS) were determined at an open-circuit voltage, with a bias of 10 mV for frequencies ranging from 100 kHz to 0.01 Hz, to test the electron transport properties.

## 3. Results and Discussion

### 3.1. The Preparation of Activated Carbon from Seaweed

Figure 1 shows the isotherm adsorption/desorption curves of activated carbon prepared by different weight ratios of seaweed/NaCl. The results show that the quality adsorbed N_2_ volume increase with the increases of the weight ratio of NaCl/seaweed, that is, the specific surface area increased with the increase of NaCl fraction; but when the addition fraction of NaCl is higher than a certain amount, the increase of the quality adsorbed N_2_ volume decreases, i.e., the specific surface area, will gradually slow down or even decrease. This phenomenon is attributed to that when the addition of weight ratio of seaweed/NaCl is 1:2~1:4, the water in NaCl aqueous solution is removed during the carbonization process, and NaCl gradually crystallizes on the algae to form a support structure template. However, when the addition of weight ratio is higher than 1:4, the quality adsorbed N_2_ volume (specific surface area) of the prepared activated carbon decreases, because when water in the NaCl solution is evaporated, too many NaCl crystals grow on the algae. It is destroyed of the framework as seaweed carbonized and converted into activated carbon. The microporous structure of carbon converted into a mesoporous microstructure.

The carbonization temperature has a significant influence on the degree of graphitization of seaweed. After the carbonization of seaweed, the graphitization of the sample can be examined by Raman spectroscopy, as shown in Figure 2. It indicates that all samples appear two distinct Raman shift characteristic peaks located at 1335 cm^−1^ and 1595 cm^−1^, the D band and G band of carbon materials, respectively. The D band is a disorder feature in carbon, and the G band indicates the state of graphitic carbon. The intensity ratio of D band to G band (I_D_/I_G_) is generally considered to a key factor of the amorphous composition in carbon materials, and a higher ratio means a lower degree of graphitization [42,43]. It can be seen from this figure that as the carbonization temperature increases from 500 °C to 900 °C, the ratio of I_D_/I_G_ decreases, indicating that the degree of graphitization increases. In preliminary studies, the carbonization temperature was as high as 900 °C. Carbonized at 900 °C, the as-prepared AC has the largest graphitized structure (the higher the temperature, the higher the degree of graphitization). However, the melting point of NaCl template is about 800 °C, and carbonization at 900 °C leads to the collapse of the prepared porous carbon structure and lead to poor pore properties. Therefore, in the Box–Behnken design in the Section 3.2, the carbonization temperature is up to 800 °C.

### 3.2. Design of Experiment (DOE) for Carbonization of Seaweed

In this study, the Box–Behnken design of the response surface methodology (RSM) was used to find out the carbonization of algae converted into activated carbon, and the process was used to optimize the specific capacitance of the porous carbon electrode for supercapacitors. The preliminary experiments showed that under the conditions of a fixed nitrogen flow rate of 50 mL min^−1^ and a heating rate of 10 °C min^−1^, the important experimental preparatory factors were carbonization temperature (600~800 °C), temperature holding time (30~90 min), the weight ratio of seaweed to NaCl (1:3~1:5), and weight ratio of water to seaweed (7.5~12.5), preparing a hieratical porous carbon material for the specific capacitance (C_s_) of the electrode.

The model analysis is shown in Table 3. The carbonization of seaweed was analyzed using four models, respectively. It can be seen that the quadratic versus 2FI model had the best “variation/total variation” (R^2^), indicating that the model can reasonably analyze the entire experimental results. For “lack of fit”, the *p*-value obtained is 0.0269 less than 0.05. When the *p* value is 0.05, representing that 5% of the experiments do not conform to the theoretical value, and the test result in this mode is insignificant. Sum of squares is used to measure how well the predicted model closes to the experimental results. The larger the sum of squares, the better of the mode applied. Among the four models, the sum of squares of quadratic vs. 2FI is the largest, indicating that the predicted results of the model match the experimental results. The best fit, so the data analysis fit the Quadratic vs. 2FI model.

Table 4 shows the analysis of variance (ANOVA) results of the second-order model. The results show that the *p*-value of the model is 0.0126 < 0.05, indicating that this model has a significant impact on the response value [44]. The *p*-values of items A, C, D, AC, AD and A^2^ in the table are all less than 0.05, indicating each item of them has a significant contribution to the regression model. Particularly, C-factor has a *p*-value of 0.0073, is the smallest of all modality factors, meaning that C (weight ratio of NaCl/SW) is the significant factor to influence the electrochemical results. After model regression analysis, a linear regression model composed of various action factors was constructed using Design Expert software (version 13), and a second-order polynomial equation can be obtained. The equation of specific capacitance calculated (C_sc_)are as follows:(2)$$C_{\mathrm{sc}}=-482.37372+2.01413\times A+0.079722\times B+54.20833\times C-48.333\times D\;-0.039\times \mathrm{AC}\;+0.0812\times \mathrm{AD}-2.34\times \mathrm{CD}-0.001933\times A^{2}\;$$

A is the carbonization temperature (°C), B is the carbonization time (min), C is the weight ratio of NaCl/seaweed, and D is the weight ratio of water/algae. The model’s assumptions of the carbonization study were validated by examining the model’s normality, equal variance, and independence.

Figure 3 shows the statistical properties of the obtained capacitance values from Box–Behnken design using Design Expert software. The normal probability plot of the residuals is shown as Figure 3a. It can be seen from the figure that the residuals of some experiments are slightly skewed, but the overall graph is linear close to 45°, and there are no outliers; therefore, it can be judged that the model does not violate the normality assumption. Figure 3b shows the scatter analysis of residuals versus predicted values. The residuals are used to analyze the scatter plot of the predicted values. If the plot has no certain shape and is not diverging or converging, the assumption of equal variance in the residuals is met. The results show that the test points are randomly distributed, and no unusual distribution appears, that the assumption of equal variance of the model holds. In Figure 3c, it is observed that the residuals are randomly distributed, without a series of positive or negative residuals; thus, the independence of the model can be determined. Figure 3d–g is the residual independence examination of each factor regression model. The independence between the error terms is judged; the scatter plot is used to test the independence of the residuals and the factors. If the random distribution is in a curved surface or a saddle-horse shape, it means that the independence is poor [45]. The distributions presented are all within the allowable range of residuals and are rectangular in shape, that the experimental variables showed an obvious independent error.

Using Design expert (version 13) for analysis, the optimal condition parameters can be found, which can effectively reduce the number of experiments and costs while improving product quality to prepare high-quality of porous carbon electrode.

In this study, the four-factor parameter was used to find the highest value of specific capacitance (Table 2). Table 5 shows the optimal conditions after software analysis. A, B, C, and D are 740 °C, 90 min, weight ratio of NaCl/seaweed is 4:1, and weight ratio of water/seaweed is 12.5, respectively, which are the optimal parameters. The specific capacitance of the porous activated carbon electrode is estimated to be 103.3 F g^−1^. The actual detected value is 110.8 (±2.0) F g^−1^.

### 3.3. Physical Property Analysis of Activated Carbon

The specific surface area and porosity of activated carbon are important properties for supercapacitor applications. The mesopores in the material can shorten the transfer path of ions and the micropores can improve the ion storage capacity. Therefore, the ratio of specific surface area and pore fraction of the material are important indicators for judging the electrochemical performance of supercapacitors [46]. In order to explore the difference in the specific surface area and porosity of activated carbon after seaweed activation, the BET specific surface area analyzer was used for revealing the specific surface property and porosity of the samples, run 2, run 15, run 20, run 21, and the optimal sample. Figure 4 shows the nitrogen adsorption and desorption curves of the above five materials. It can be observed that there is not much difference in the types of adsorption and desorption curves of each material (Figure 4a). Comparing the hysteresis loops shapes of the samples from Figure 4a, it can be seen that the samples have similar porous microstructures. All five samples belong to the type IV adsorption isotherm. The curves show the shape of large opening type hysteresis loop of H2, indicating that the material contains a wide range of pore sizes and shapes. However, it shows that the hysteresis loop of the optimal sample is largest, meaning that the pore and pore volume of the material are largest. At the same relative pressure, the adsorbed gas (N_2_) amount is the largest, showing the largest specific surface area. The hysteresis loops at medium relative pressures indicate the presence of mesoporous and conform the hierarchical micro/mesoporous nature of the as-obtained carbon derived from seaweed [47].

Table 6 is the analysis of the specific surface area and pore fraction of the materials. Furthermore, the optimal activated carbon sample has the most adsorption capacity and possesses the highest specific surface area. Among them, although the amount of sodium chloride added in the run 20 sample is the same as that of the run 21 sample, however; the activation temperature of run 20 is 800 °C, which is about the melting point of sodium chloride that it (run 20) cannot be used as a template support structure, because the microstructure collapses during the carbonization at the temperature. The specific surface area decreased.

Figure 4b shows the pore size distribution of the materials. The results show that most of the pore sizes of the materials are between 1 and 100 nm. According to the IUPAC classification, pores are divided into three types: micropores with a size less than 2 nm; macropores with a size greater than 50 nm; mesopores with a size between 2 nm and 50 nm; thus, it can be judged that the activated carbon synthesized in this study composed of micropores and mesopores, which is consistent with the above isotherm adsorption curve. The activated carbon prepared under the optimized conditions has the highest specific surface area of 603 m^2^ g^−1^; and the specific surface area of the other four samples increased with the addition of sodium chloride except the sample run 20.

From the BET analysis, it can be speculated that the pore proppant of NaCl crystals introduced into the seaweed effectively supports the bulk carbon material against framework shrinkage. After the NaCl is dissolved and removed, large range sizes of pores are left in the final sample and promoting the formation of hierarchical porous carbon [48]. The use of NaCl as a template for the carbonization of seaweed is attributed to the unique porous structure of the prepared samples. The porous carbon material with a large range of micropores and mesopores in the material can increase the contact area of the electrolyte in the material, which is conducive to penetration into the electrode, thereby improving the transmission of ions and the energy storage of supercapacitors.

Scanning electron microscope (SEM) was utilized to reveal the surface morphology of activated carbon samples as shown in Figure 5. Different magnifications of SEM images are also provided on Appendix A. Obviously, there are many pores of different sizes existed in the different samples. Combining with the BET study, as the specific surface area increases, the pore structure also increases. Theoretically, increasing the pores can improve the ion diffusion rate and ion storage capacity.

In the application of activated carbon for supercapacitor, the degree of graphitization of activated carbon is also an important indicator to improve the electrical properties. The degree of graphitization of the sample can be identified by the I_D_/I_G_-value of Raman shift as shown in Figure 6a. As seeing from the figure that as the carbonization temperature of different samples increases, the degree of graphitization is also higher. Among these shown samples, run 20 had the lowest I_D_/I_G_ value of 0.986 under the activation condition of 800 °C, which represented the highest degree of graphitization. In theory, the sample run 20 has a highly graphitized structure that should exhibit excellent capacitive properties. However, as previously speculated, at the high temperature of 800 °C, part of the NaCl crystalline supporter collapsed due to melting of NaCl at the temperature, resulting in poor microstructural properties of activated carbon and unsatisfactory capacitance properties.

Figure 6b shows XRD patterns of the as-obtained different porous carbon. The patterns exhibit a broad diffraction peak of 2θ at around 24°, which is corresponding to the (002) reflection of graphic carbon with a disorder phase [47]. Additionally, all the samples possess a typical sharp peak at 26.5°, which can be indexed the as-prepared carbons are related to the presence of graphite phase [49]. In addition, the optimal sample and run 2 sample, it has a broader (002) peak, and the peak tends to a relative lower angle, which means that the material is mainly composed of finer amorphous carbon. Moreover, the optimal and run 20 samples show stronger sharp peaks at 26.5°. This may be attributed to the fact that these two samples were carbonized at higher temperature and exhibits higher graphitized structure. In contrast, run 2 and run 21 samples were carbonized at lower temperature, the XRD patterns show a weaker graphitic phase. The effect of carbonization temperature on graphitization is consistent with the previous results of Raman’s analysis.

### 3.4. Electrochemical Analysis of Activated Carbon

In this study, a three-electrode system was used to investigate the electrochemical properties of the prepared activated carbon electrode materials. The saturated calomel electrode (SCE) was used as the reference electrode, the platinum plate was used as the opposite electrode, and the 1 M Na_2_SO_4_ solution was used as the electrolyte. Cyclic voltammetry (CV) and galvanostatic charge–discharge (GCD) tests were performed.

The CV characteristics of the as-prepared activated carbon electrodes are shown in Figure 7. The shapes of CV curves of samples are all rectangular, indicating good EDLC characteristics.

The charge–discharge behavior of activated carbon electrodes of samples is shown in Figure 8. It indicates that each charge–discharge curve is a symmetrical triangle with good capacitance performance. Calculate the specific capacitance value using Equation (1). The specific capacitance of run 2, run 15, run 20, run 21, and optimal is 86.3, 74, 56.1, 31.1, and 110.8 F g^−1^, respectively (Figure 8a). Table 7 the properties of the as-obtained activated carbon. The micropore area of activated carbon in each sample accounts for 70~80% of the overall volume. Due to the two characteristics of mesopores shortening ion migration distance and micropores increasing ion storage capacity. Therefore, under the same pore fraction conditions, the specific capacitance of the electrode increase as the increase of the specific surface area of the activated carbon; however, due to the different degree of graphitization of the activated carbon of each sample, run 21 has a specific surface area of 507.1 m^2^ g^−1^, but the specific capacitance is only 31.1 F g ^−1^, the optimal sample has both a high specific surface area of 603.7 m^2^ g^−1^ and a high degree of graphitization of I_D_/I_G_ at 0.989; thus, it has the highest capacitance of 110.8 F g^−1^. Figure 8b shows the GCD test of the optimal sample at different current densities. It is calculated from the figure that the specific capacitance of the optimal electrode at different current densities of 1, 2, 4, 6, 8, and 10 A g^−1^ are 110.8, 85.7, 74.9, 68.6, 62.9, and 57.1 F g^−1^, respectively. This is because as the current density increases, the ions in the electrolyte only react rapidly with the surface of the material and cannot fully enter the pore structure for storage, resulting in a decrease in specific capacitance [50].

Figure 8c shows the impedance analysis of different samples of activated carbon. The Nyquist plot can be divided into two parts, including a semicircle in the high frequency region and a steep line in the low frequency region. In the low frequency region, the impedance represents the diffusion resistance for the electrolyte ions in the holes of the electrode. The optimal electrode has a straight line with very steep slope compared to other samples, indicating that the capacitance performance is very close to that of an ideal supercapacitor, owing to the faster ion diffusion and migration. In the high frequency region, it can be divided into the equivalent series resistance (ESR) of the electrode material and the interface charge transfer resistance R_ct_. It can be seen from the figure that the ESR of run 2, run 15, run 20, run 21, and optimal resistance values is 2.6 Ω, 3.2 Ω, 3.5 Ω, 2.9 Ω, and 2.0 Ω, respectively, and there is no obvious semicircle curve in the figure, which is mainly attributed to the fast ion response and low interfacial impedance in the porous structure [51].

Summing up the above analysis, it can be known that the ion diffusion efficiency of the optimal electrode is higher. After 10,000 charge/discharge cycles at a current density of 1 A g^−1^, the optimal is tested for stability. It still retains 95.4% of the initial capacitance value as shown in Figure 9.

Additionally, the optimal electrode was examined by SEM, comparing the electrode morphology before/after 10,000 charge/discharge cycles test, and no obvious cracks were found (Appendix A). Table 8 shows a comparison of electrochemical performance of carbon from other biomass in the literatures. In this study, NaCl used as the template during carbonization to obtain the optimal carbon electrode. The resulting capacitive properties of the electrode are not inferior to carbon electrodes made with other biomass/activators. Moreover, the use of NaCl will greatly reduce the pollution to the environment, which is very worthy of attention.

## 4. Conclusions

An environmentally friendly, simple and economical process was developed by utilizing seaweed as the carbon precursor. During the carbonization process, the design of experiment (DOE) technology was applied to plan the green process without using a strong acid and alkali as the activator. In contrast, NaCl used as the pore structure proppant of seaweed for carbonization to prepare a hierarchical porous carbon, which was used as the electrode material of super capacitor. In the process of seaweed carbonization, the experimental variables were activation temperature, activation time, weight ratio of NaCl/seaweed and weight ratio of water/seaweed.

It is obviously found that the specific surface area of activated carbon increased with the increase of NaCl/seaweed; however, when the addition ratio is higher than 1/4, the increasing trend of the specific surface area will gradually slow down or even decrease. When the activation temperature was increased from 500 °C to 900 °C, the Raman shift intensity ratio (I_D_/I_G_) between the D band and the G band decreased, indicating an increase in the degree of graphitization.

In addition, surface response methods (including Box–Behnken design) was used to optimize the process for carbonization of seaweed. It revealed that the optimal preparatory conditions were activation temperature at 740 °C, activation time at 90 min, weight ratio of NaCl/seaweed at 4 and weight ratio of water/seaweed at 12.5. A hieratical porous activated carbon electrode was prepared with a maximum specific surface area of 603.7 m^2^ g^−1^ and a maximum capacitance of 110.8 F g^−1^. At a current density of 1 A g^−1^, the material still retains 95.4% of the initial capacitance after 10,000 cycles of stability testing. Using the DOE to plan the green process of using NaCl as the carbon material structural template (template), the prepared hierarchical porous carbon material has better energy storage properties than the supercapacitor made of traditional carbon material.

## Data Availability

Data sharing is not applicable for this article.

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
