# Peer review of "Preparatory Conditions Optimization and Characterization of Hierarchical Porous Carbon from Seaweed as Carbon-Precursor Using a Box—Behnken Design for Application of Supercapacitor"

_materials, 2022, doi:10.3390/ma15165748_

Round 1

Reviewer 1 Report

Dear Authors

This manuscript is focused on the development an environment-friendly, simple and economical process by utilizing seaweed as a carbon precursor to prepare a hierarchical porous carbon for the application of supercapacitor. The following suggestion and comments should be taken:

1. The overall English needs to be improved. Please seek guidance from a native English speaker if possible ("the" "a", commas, plural form and others could be corrected).

2. The introduction section needs enhancement about different applications of porous activated carbons with the use of different activators. Please cite (in lines 43-46) (1) Processes 2022, 10(7), 1359; https://doi.org/10.3390/pr10071359 (2) Materials 12 (20), 3354, 2019 https://doi.org/10.3390/ma12203354 (3) Nanomaterials 2021, 11(9), 2217; https://doi.org/10.3390/nano11092217

3. Schematic A, please correct this image for better quality (the inscriptions on the drawing).

4. Could the authors include the standard deviation of the used methods?

5. Figure 2. Please correct this image for better quality.

6. Figure 3. Please correct this image for better quality (the inscriptions on the drawing).

7. Figure 4. Please correct this image for better quality.

8. What authors can say about visible hysteresis loops in Figure 4?

9. Figure 5 please add SEM images with different magnifications

10. Why do authors choose only one magnification for SEM?

11. Figure 6. I think this figure is obvious. As the temperature increases, the degree of graphitization increases. Maybe for supplementary information.

12. In the electrochemical part (Cv curves) - why do the authors use a scan rate of 10-50 mV s-1. Can you explain that?

13. Which parameters of materials influence on the electrochemical results the most?

14. Generally, for supercapacitors, the sharper the angle is the better for the results. Please explain the result for 1 A/g?

Reviewer 2 Report

In this work, the authors used seaweed as carbon precursor to prepare a hierarchical porous carbon for application of supercapacitor. Preparation of porous carbon from seaweed for use in supercapacitors has been reported. In addition, there are several inconsistencies and the work is not well organized, so I do not recommend publication. The comments are as follows:

1.       The authors claimed to use NaCl as an intercalating agent, but used sodium hydroxide in the description of the Experimental.

2.       The authors did not list 900 °C when designing the experiment (Table 1). But 900 °C came up in the resulting discussion. The authors are confused about the design of the experiment.

3.       In the electrochemical test, the author used MnO2/graphene/Ni foam composite as the working electrode, then what is the use of carbon prepared by seaweed?

Reviewer 3 Report

This paper reports the fabrication of hierarchical porous carbon materials from seaweed via carbonization with the Box-Behnken design. The prepared hierarchical porous carbon materials have been characterized by various techniques, such as BET, SEM and Raman spectroscopy. Using DOE method, the authors were able to find the optimal conditions for activating the seaweed and subsequently, optimizing the electrochemical performance for supercapacitors, leading to an optimal sample with specific capacitance of 110 F/g at a current density of 1 A/g and good capacitance retention of 95.4%. Overall, this work has demonstrated the use of Box-Behnken as a valuable tool for optimising the electrochemical properties of seaweed-derived hierarchical porous carbon. Therefore,  this work will be of interest to the readers. However, some revisions are needed before this work can be accepted as detailed below:

1.  XRD characterization of run 2, run 15, run 20, run 21, and the optimal sample should be provided and discussed.

2. How about the morphology of the optimal electrode after long cycling? Is it still preserved or changed?

3. In Fig. 5, the font size of the scale bars is not consistent (the same). Please fix this.

4. The role of NaCl in promoting the formation of hierarchical porous carbon needs to be explained in more details.

5. A Table comparing the electrochemical performance of the optimal electrode with other carbon-based materials need to be provided.

6. In the Introduction, some recent references on the development of different types of carbon-based electrode materials, such as  Electrochimica Acta, 394, 139058 (2021); Chemical Communications 58 (7), 1009-1012 (2022) and Chemistry of Materials 30 (13), 4401-4408 (2018)  can be mentioned and cited to provide a broader perspective of the variety of carbon-based electrode materials for supercapacitors.

Round 2

Reviewer 1 Report

The authors have addressed all comments and the manuscript can be published as is.

Reviewer 2 Report

The author has revised my questions in detail, but in the Introduction, the application of biological carbon in other fields can be added, such as literature: Adv. Energy Mater. 2021, 11, 2003699; And the latest progress of micro supercapacitors: eScience 1 (2021) 124-140.
